# An exploration of the journey to diagnosis of Ehlers-Danlos Syndrome (EDS) for women living in Australia

Rachel Trudgian[1], Terri Flood[2]*

1 Skills for Development Pty Ltd., Clear Mountain, Queensland, Australia, 2 School of Health Sciences, Ulster University, Derry~Londonderry, United Kingdom

* t.flood@ulster.ac.uk

## Abstract

### Introduction

Ehlers-Danlos syndromes (EDS) is an umbrella term for a group of hereditary connective tissue disorders usually featuring hyperextensible skin, easy bruising, poor healing, and joint hypermobility. According to international Ehlers-Danlos support groups, the average time to diagnosis of this rare genetic condition is 10–12 years. Consequently, the journey to diagnosis can potentially be traumatic. This pilot study aims to explore female patients' journey to a diagnosis of EDS while living in Australia.

### Materials and methods

Over six weeks, from January to February 2023, a survey was distributed on EDS social media platforms including Facebook and LinkedIn. Ethical approval was obtained through the Ulster University Institute of Nursing and Health Research Ethics Filter Committee (FCNUR). Thematic and statistical analysis of the anonymous data was completed during March 2023.

### Results

152 women completed the survey. More than half of the respondents first noticed symptoms of EDS more than 15 years prior to diagnosis and more than three quarters of respondents received other diagnoses prior to their EDS diagnosis. Most respondents saw their general practitioner, a physiotherapist and/or a rheumatologist prior to being correctly diagnosed with the condition. While some respondents had positive experiences with these health professionals, many felt that they were not listened to and, after doing their own research, had to educate their health professional/s about EDS.

### Conclusion

This pilot survey demonstrated that the journey to diagnosis of EDS for women in Australia is frequently long and traumatic. Participants indicated that more EDS education and training is needed for health professionals, especially GPs, to improve the diagnostic process.

**Data Availability Statement:** Access to the full data can be obtained by request to the Nursing and Health Research Ethics Filter Committee in Ulster University (contact wg.kernohan@ulster.ac.uk).

**Funding:** The author(s) received no specific funding for this work.

**Competing interests:** The authors have declared that no competing interests exist.

## Implications for practice

Participants called for their self-reported symptoms to be listened to by health professionals and to be met with empathy and an open mind instead of being dismissed. Additional education and training to health professionals about connective tissue disorders including EDS may aid earlier diagnosis in Australia.

## Introduction

Ehlers-Danlos syndromes (EDS) are a group of rare heritable connective tissue disorders [1], characterised 'by the abnormal formation and/or assembly of collagen, fibrillin and elastin in the body' (page 1) [2]. EDS can lead to multi-systemic complications including generalised joint hypermobility, joint instability complications, widespread musculoskeletal pain, poor skin integrity, cardiovascular dysfunction, and gastrointestinal dysfunction [2]. There is no cure for EDS [1,3], only symptom management and prevention of further deterioration [4]. While historically the prevalence of EDS was thought to be approximately 1 in 5000 cases annually worldwide, more recent figures suggest that prevalence is higher at around 120 per 100,000 [5].

Based on the 2017 international classification system, EDS is divided into 13 subtypes [1,6,7] of which Hypermobile EDS (hEDS) is the most common type [8,9], estimated to account for approximately 90% of EDS diagnoses [10]. hEDS is generally considered to be the least severe type of EDS but can lead to significant musculoskeletal complications resulting from hypermobility and joint instability, along with other comorbid conditions [10,11]. While skin hyperextensibility is seen in many types of EDS, this symptom is less common in hEDS [10]. Symptom severity and progression of hEDS is likely impacted by factors such as age, gender and lifestyle [7]. While most types of EDS are caused by mutations in genes, no gene has yet been identified as the cause of hEDS [1]. Consequently, a genetic diagnosis is indicated for all subtypes, except hEDS [10].

Past challenges in diagnosing hEDS have included poor knowledge of medical professionals due to its rarity [12–16], its multisystem nature [10,17,18] and the lack of consistent diagnostic criteria for hEDS [16]. New criteria was developed in 2017 by the EDS International Consortium, to provide consistent diagnostic criteria with the aim of reducing future misdiagnoses [10]. Despite this change in diagnostic criteria, recent studies by Halverson et al. (2021) [13] and Wang et al. (2024) [14] have reported that the average time to diagnosis of hypermobility spectrum disorders (HSD), including Ehlers-Danlos syndrome, to be 11–16 years. Participants consistently report being referred to numerous specialists and obtaining multiple alternative diagnoses prior to their formal hEDS diagnosis [12–14,19]. As a result, many hypermobile individuals are not correctly diagnosed until much later in life when joint damage may be irreversible [12]. Misdiagnoses can lead to inappropriate and sometimes dangerous treatment of their presenting symptoms [20], potentially resulting in an altered sense of self [21]. Earlier diagnosis is vital to ensure appropriate treatment and prevent future complications [22]. Obtaining a diagnosis of hEDS often then opens doors for obtaining additional, accurate diagnoses of postural orthostatic tachycardia Syndrome (POTS) and mast cell activation syndrome (MCAS) [23]. This is important as patients are increasingly presenting with this triad of syndromes together [9]. Recent studies which have assessed the lived experience of people through their diagnosis of EDS, have included experiences from the United Kingdom, Sweden and Belgium [12] and the United States of America [12–14]. The lived experiences of people receiving

a diagnosis of EDS in Australia is largely unreported. Given that Australia's healthcare system was independently ranked 5[th] in the world in 2022, above all of these countries [24], it is important to gain an understanding of the EDS diagnostic experience to add to the global evidence of this rare disease.

Several reports indicate that hEDS occurs predominantly in females with over 90% of confirmed cases of hEDS being female [5,9,25]. Recent studies indicate that women with hEDS symptoms have a reduced quality of life and experience more severe sexual dysfunction (often linked to pelvic organ prolapse) compared to men [26]. Additionally, suicide risk is high in hEDS, with women reporting higher rates of suicide attempts compared to men [26]. Baeza-Velasco et al.'s 2022 study of women with hEDS reported that 31.4% of the sample had made at least one suicide attempt with 42.8% of patients assessed indicating some risk of suicide [27]. Given that the average time to diagnosis of hEDS is 10–16 years [12–14] it is essential that information is collated from women who have been through this process to truly understand their lived experiences. Further, due to the lack of a pathognomonic biomarker for hEDS, patient reports of their symptoms, particularly before they are accurately diagnosed with hEDS, are often dismissed [12–14,28]. Patients are sometimes referred for psychiatric evaluation, often told it is 'all in their head' and not real and classified as 'somatisers' [12–14,29]. A recent Gender Pain Gap Index report [30] in the UK indicated that dismissal of physical symptoms most frequently occurs to women. For the same pain symptoms, only 44% of women surveyed received a diagnosis for their pain compared to 66% of men within 11 months [30]. Thirty percent of women felt that their time to diagnosis was negatively impacted because of their health professional dismissing their pain compared to 18% of men [30].

The primary aim of this study was to capture female patients' perspectives and experience regarding their journey to diagnosis of EDS while living in Australia. Factors explored which may influence time to diagnosis included having other family members with an EDS diagnosis, types of health professionals visited and diagnostic tests used. Understanding factors which can impact time to an EDS diagnosis can help to highlight changes needed to the EDS diagnostic pathway and/or support services required for this rare condition in Australia.

## Materials and methods

### Study design

A pilot study was deemed appropriate as the first step to provide initial impressions of the diagnostic experience and potential interventions to help design future experimental research in this topic area [31].

A survey was chosen as the data collection method as it allows for data to be collected from a large, targeted population [32]. Given the large geographical distance between cities in Australia, a survey was deemed the most appropriate method. JISC Online Survey (formerly Bristol Online Survey—BOS) was utilised to distribute the survey anonymously and was supplied free of charge as part of a departmental contract with Ulster University. Due to fluctuating COVID restrictions at the time, and considering that many participants may be immunocompromised, an online survey had the advantage of enabling participants to complete the survey in the comfort of their own homes. Participants were also able to opt out of answering any question if they wished. A combination of open and closed response questions were chosen to allow flexibility and depth of responses. JISC enabled respondents to save their place in the survey and return at another time, provided it was within the six-week period. This was important, particularly given the levels of fatigue and pain respondents may have experienced while accessing an online survey, secondary to their EDS. Initial question design was based on an extensive literature search and the lead researcher's anecdotal experience as an occupational

therapist. Content validity was increased through discussions between the two researchers and following discussions with women with EDS living in Brisbane, Australia [33]. These discussions resulted in some minor modifications to the questionnaire; this was primarily related to ensuring the use of inclusive language throughout the survey. Questions were arranged in a logical order collecting demographic information in the earlier questions of the survey and then asking about participants' experiences before, during and after they received their diagnosis of EDS. See survey in supplemental information. See Supporting information for full survey.

### Participant recruitment

People were eligible to participate in this study if they identified as female, were over the age of eighteen, and had been living in Australia for more than one year before obtaining their diagnosis of EDS in Australia.

Participants were recruited by convenience sampling via social media (Facebook and LinkedIn). A review of all open and accessible Facebook pages relevant to the EDS community in Australia was conducted. Eleven Facebook pages were identified. Due to rules of the individual pages (no surveys allowed; only individuals with EDS were allowed to join), seven pages accepted the survey link to be shared. Of those seven pages, membership totalled 8,200, though there would likely have been crossover between the different pages as members are free to join as many pages as they wish. The survey was advertised by the lead author on the day the survey opened, 3rd January 2023, then reminders were posted at 2-week intervals and on the second last day before the survey closed on 15th February 2023.

### Consent and ethics

Ethical approval for this study was obtained through the Nursing and Health Research Ethics Filter Committee in Ulster University in September 2022. The approval number for this study is FCNUR-22-062-A. The lead author conducted this research as part of an MSc through Ulster University, while practicing as an occupational therapist in Brisbane, Australia.

A brief cover letter accompanying the link to the survey outlined the inclusion criteria and the closing date of the survey. Participants were encouraged to click on the survey link and advised that the survey could take up to 20 minutes to complete. The first page of the survey outlined the purpose of the survey and invited participants who met eligibility criteria to continue. Participants had to indicate that they met each of the eligibility criteria to continue into the survey. It was clearly stated that the survey was anonymous and that participation in the survey implied consent by virtue of survey completion [34]. Participants were also informed, prior to starting the survey that the survey discussed potentially sensitive topics, which may cause some distress and subsequently that they may withdraw at any point until their survey is submitted.

Once the survey had been live for six weeks, it automatically closed and participants were no longer able to access the survey. The lead author posted a statement of thanks on the seven Facebook pages and LinkedIn page to the women who completed the survey.

### Sample and sampling

Globally, as previously highlighted, EDS is thought to occur in approximately 120 per 100,000 people [35,36]. Based on this figure, and Australia's population of over 26 million [37], there are approximately thirty-one thousand people living with various types of EDS in Australia, the majority of which are female [5,9,25]. An acceptable margin of error used in survey research is 5–10%, with a confidence level of 95% and a 50% response distribution [38]. Using

the Raosoft sample size calculator [39], it was, therefore, determined that between 96 and 380 participants would need to be successfully recruited in order to yield statistically significant results.

### Data analysis

Descriptive statistics were used to analyse data for single categorical variables and included frequencies and percentages. Cross tabulation was conducted to quantitatively analyse the relationship between variables including time to diagnosis, time of diagnosis and family history of EDS.

Qualitative analysis of the data was conducted using inductive thematic analysis as described by Williams and Moser [40]. Inductive analysis involves deriving meaning from the data without preconceptions [40]. Participants' qualitative responses were captured through JISC and exported into an excel spreadsheet for coding. The first stage of coding included open coding; this involved read and re-reading of the responses to increase familiarity and understanding of the data [40]. During this stage sequences of words or phrases were manually coded in an iterative process. Unlike full interviews or focus groups, the volume of data was relatively low and therefore specialist software for aiding data analysis was not deemed necessary. Code notes were produced to explain the content of each of the codes. Axial coding was the second stage of coding in this analysis and involved development of the codes into categories and themes by identifying relationships between open codes [40]. This process was dynamic, requiring consideration of possible influencers relative to the findings.

Both study researchers independently completed the open coding and axial coding stages of this process. To improve inter-rater reliability, the researchers shared their code notes with each other [41] and discussed any discrepancies between the coding. Through reflection, reflexivity and discussion in a non-linear process, both researchers agreed on all final codes, categories and themes, exceeding suggested inter-rater reliability of 80% agreement between coders on 95% of the codes. This process of independent coding addresses the issue of objectivity and increases the rigour of the data analysis process [42].

## Results

One hundred and fifty two eligible participants completed this survey regarding their journey to a diagnosis of EDS in Australia.

### Demographic information

More than half (n = 79; 52%) of participants indicated that they lived in Queensland. The majority of participants identified that they were between 25 and 44 years old (n = 83; 54.6%) with 46 participants (30.3%) indicating that they were 45 years or older.

Participants most frequently identified that their highest level of education was completion of an undergraduate degree at university (n = 48; 31.6%), while 36 respondents (23.7%) had completed postgraduate studies. Sixty-five participants (32.8%) reported that they were working either full-time, part-time or on a casual basis. Some participants indicated that they were currently studying (n = 17; 11.2%). Fifteen (9.9%) respondents indicated that they are unemployed while 37 (24.3%) participants indicated that they are currently receiving the Disability Support Pension. See Table 1 for full demographic information.

**Table 1. Demographic information of participants.**

| State of residence | n | (%) |
|---|---|---|
| Queensland | 79 | 52.0 |
| Victoria | 37 | 24.3 |
| New South Wales | 24 | 15.8 |
| South Australia | 4 | 2.6 |
| Western Australia | 4 | 2.6 |
| Tasmania | 3 | 2.0 |
| ACT | 1 | 0.7 |
| Northern Territory | 0 | 0.0 |
| Age range | n | (%) |
| 18–24 | 23 | 15.1 |
| 25–34 | 40 | 26.3 |
| 35–44 | 43 | 28.3 |
| 45–54 | 31 | 20.4 |
| 55–64 | 12 | 7.9 |
| 65+ | 3 | 2.0 |
| Educational level | n | (%) |
| Undergraduate degrees | 48 | 31.6 |
| TAFE/college | 47 | 30.9 |
| Postgraduate studies | 36 | 23.7 |
| Finished high school (Year 12) | 25 | 16.4 |
| Finished school before Year 12 | 12 | 7.9 |
| Employment Status | n | (%) |
| Working full-time | 24 | 15.8 |
| Working part-time | 28 | 18.4 |
| Working casually | 13 | 8.6 |
| Homemaker | 15 | 9.9 |
| Retired | 6 | 3.9 |
| Studying | 17 | 11.2 |
| Business owner | 11 | 7.2 |
| Unemployed | 15 | 9.9 |
| Receiving DSP | 37 | 24.3 |
| Other | 27 | 17.8 |

Note: Participants were able to select more than one option for Educational level and Employment Status.

## Types of EDS

There are thirteen subtypes of EDS, of which hypermobile EDS (hEDS) is the most common [35]. The majority of participants (n = 137; 90.1%) identified as having Hypermobile EDS (hEDS) which is consistent with previous research [1,43]. See Table 2 for full details.

Forty-three respondents (28.3%) had a family member who had been diagnosed with EDS in Australia while 109 respondents (71.7%) were the only members of their family diagnosed with EDS. Of the 43 respondents who had a family member diagnosed with EDS, 32.6% indicated that this was their daughter. See Table 3 for full details.

Cross tabulation revealed that, despite 43 respondents having a family member diagnosed with EDS, 32 of these respondents (74.4%) had symptoms for more than 15 years prior to obtaining an EDS diagnosis.

**Table 2. Types of EDS.**

| Type of EDS | n | (%) |
|---|---|---|
| Hypermobile EDS (hEDS) | 137 | 90.1 |
| Unknown | 6 | 3.9 |
| Classical EDS | 3 | 2.0 |
| Classical-like EDS | 3 | 2.0 |
| Vascular EDS (vEDS) | 2 | 1.3 |
| Cardiac-valvular EDS (cvEDS) | 1 | 0.7 |

Thirty-seven participants (86%) who had a relative with an EDS diagnosis, took 10 years or greater to be diagnosed with EDS compared to 78 (71.6%) of those who did not have a relative with a diagnosis. See Table 4.

## Prior to diagnosis of EDS

The majority of participants (n = 96; 63.2%) had been diagnosed with EDS since 2020. Fifty-three participants (34.9%) were diagnosed between 2010 and 2019. Ninety-six participants

**Table 3. Family members diagnosed with EDS.**

| | N | (%) |
|---|---|---|
| Participants who did not have a family member diagnosed with EDS | 109 | 71.7 |
| Participants with family member diagnosed with EDS | 43 | 28.3 |
| Family member diagnosed with EDS; | | |
| • Daughter | 14 | 32.6 |
| • Sister | 9 | 20.9 |
| • Son | 9 | 20.9 |
| • Mother | 8 | 18.6 |
| • Brother | 1 | 2.3 |
| • Grandmother | 1 | 2.3 |
| • Father | 1 | 2.3 |

**Table 4. Participants indicate how long they had symptoms prior to obtaining their official diagnosis (Cross tabulation of question 5 and question 7).**

| How long prior to obtaining your official diagnosis had you noticed symptoms? | Do you have a family member who has been diagnosed with EDS in Australia? | | Totals |
|---|---|---|---|
| | Yes | No | |
| Less than 1 year | 2 | 2 | 4 |
| 1–2 years | 1 | 6 | 7 |
| 3–4 years | 1 | 13 | 14 |
| 5–9 years | 2 | 10 | 12 |
| 10–14 years | 5 | 14 | 19 |
| More than 15 years ago | 32 | 64 | 96 |
| No answer | 0 | 0 | 0 |
| **Totals** | **43** | **109** | **152** |

(63.2%) first noticed symptoms of EDS more than 15 years prior to their diagnosis. Prior to receiving their confirmed diagnosis of EDS, most patients saw their general practitioner (n = 144, 94.7%), a physiotherapist (n = 131, 86.2%) and/or a rheumatologist (n = 101; 66.4%). Other specialists consulted before being formally diagnosed with EDS included gastroenterologists, orthopaedic surgeons, gynaecologists, pain specialists, and neurologists/neurosurgeons. Other allied health professionals who were consulted included osteopaths, psychologists and occupational therapists.

The majority of participants (n = 129; 84.9%) indicated that they received alternative diagnoses prior to their formal EDS diagnosis. Most participants reported being diagnosed with anxiety and/or depression (n = 91; 73.4%) or another psychological condition (n = 47; 37.9%), chronic pain (n = 67; 54%), myalgic encephalomyelitis/chronic fatigue syndrome (ME/CFS) (n = 58; 46.8%), or another physical condition including fibromyalgia (n = 18; 17.7%). Over 10% of respondents (n = 14) reported receiving more than nine diagnoses before being diagnosed with EDS. See Table 5 for further details.

Cross tabulation revealed that 72 out of the 96 participants (75%) who were most recently diagnosed in the period from 2020 to present, had symptoms of EDS for over 15 years before their official diagnoses. See Table 6.

## After believing they had been misdiagnosed

From the 129 participants who indicated that they believed that they had received a misdiagnosis, 125 (96.9%) responded with comments to the question asking how they felt after being misdiagnosed. All of these participants identified that they felt negative emotions with some participants expressing more than one negative emotion. These emotions were stratified into two themes; 1) Feelings towards the Health Professional involved and 2) Internal feelings. The most common emotions expressed regarding the health professionals involved in their care included feeling frustrated/angry/annoyed (n = 50; 40%), ignored/dismissed/not taken seriously (n = 30; 24%) and patronised (treated like a hypochondriac) (n = 26; 20.8%). The most common internal emotions described included feelings of hopeless (n = 25; 20%), and feeling sad/depressed (n = 17; 13.6%) after being misdiagnosed. See Table 7 for further coding information.

Participants explain their feelings:

'Frustrated. I knew there was something more to my symptoms.' (Participant 1) 'Hurt, not seen, mistreated, angry, confused, lonely, depressed.' (Participant 3) 'Felt no hope. Told it was all 'in my head' a lot.'

(Participant 9)

'Gaslit–like I was being dramatic, lazy, less than, making me question my identity and my experiences (especially being misdiagnosed as having trauma and I'd repressed it).'

(Participant 50)

Recurring themes when participants had negative experiences with health professionals included not being listened to/believed, being dismissed, or made to feel 'stupid'; that their symptoms were all in their head.

One hundred and fourteen participants (75%) sought self-diagnosis online with 104 of these participants (91.2%) identifying that the information available online was helpful. Of the 104 participants who found online information helpful, the resources most valued were

**Table 5. Experience prior to a diagnosis of EDS.**

| Obtained diagnosis of EDS | n | (%) |
|---|---|---|
| 2020–present | 96 | 63.2 |
| 2010–2019 | 53 | 34.9 |
| 2000–2009 | 2 | 1.3 |
| Before 2000 | 1 | 0.7 |
| **Noticed symptoms of EDS before official diagnosis** | **n** | **(%)** |
| Less than 1 year | 4 | 2.6 |
| 1–2 years | 7 | 4.6 |
| 3–4 years | 14 | 9.2 |
| 5–9 years | 12 | 7.9 |
| 10–14 years | 19 | 12.5 |
| More than 15 years | 96 | 63.2 |
| **Professional consulted before formal diagnosis** | **n** | **(%)** |
| General practitioner | 144 | 94.7 |
| Physiotherapist | 131 | 86.2 |
| Rheumatologist | 101 | 66.4 |
| Other | 71 | 46.7 |
| Cardiologist | 60 | 39.5 |
| Immunologist | 29 | 19.1 |
| **Were you misdiagnosed prior to your diagnosis of EDS?** | **N** | **(%)** |
| Misdiagnosed | 129 | 84.9 |
| Correctly diagnosed | 23 | 15.1 |
| **Number of misdiagnoses** | **n** | **(%)** |
| 1–3 | 73 | 56.6 |
| 4–6 | 36 | 27.9 |
| 7–9 | 6 | 4.7 |
| More than 9 | 14 | 10.9 |
| **Misdiagnosis*** | **n** | **(%)** |
| Anxiety and/or depression | 91 | 73.4 |
| Chronic pain | 67 | 54.0 |
| Other physical | 58 | 46.8 |
| Generalised joint laxity | 52 | 41.9 |
| Other psychological | 47 | 37.9 |
| Chronic fatigue syndrome (CFS) | 45 | 36.3 |
| Dysautonomia | 18 | 14.5 |
| Fibromyalgia | 18 | 17.7 |
| Myalgic encephalomyelitis (ME) | 13 | 10.5 |
| Rheumatoid/Osteo/other arthritis | 6 | 5.9 |
| Marfan syndrome | 4 | 3.2 |

* Participants could select more than one misdiagnosis.

websites (n = 88; 84.6%), Facebook support groups (n = 78; 75%) and medical journal articles (n = 76; 73.1%). See Table 8 for further information.

## Formal diagnostic process

Ultimately, 54.6% of respondents were formally diagnosed by a rheumatologist (n = 83), 29.6% by a geneticist (n = 45), 8.5% by a GP (n = 13) and 6.6% by a physiotherapist (n = 10).

**Table 6. Participants indicate how long they had symptoms prior to obtaining their official diagnosis (Cross tabulation of question 6 and question 7).**

| Time period of diagnosis | Length of time in years to diagnosis of EDS | | | | | | |
|---|---|---|---|---|---|---|---|
| | **<1** | **1–2** | **3–4** | **5–9** | **10–14** | **>15** | **Total** |
| 2020-present | 3 | 2 | 6 | 4 | 9 | 72 | 96 |
| 2010–2019 | 1 | 5 | 7 | 7 | 10 | 23 | 53 |
| 2000–2009 | 0 | 0 | 1 | 1 | 0 | 0 | 2 |
| Before 2000 | 0 | 0 | 0 | 0 | 0 | 1 | 1 |
| | 4 | 7 | 14 | 12 | 19 | 96 | 152 |

During the diagnostic process, most participants were assessed using the Beighton scale (n = 136; 89.5%), self-report of pain/other symptoms scales (n = 117; 77%), clinical criteria (n = 116; 76.3%), the international diagnostic checklist for hEDS (n = 93; 61.2%) and tools to assess the presence of other potentially associated disorders (n = 88; 57.9%). See Table 9 for further information.

One hundred and forty six of the 152 respondents (96.1%) included comments about how they felt during the diagnostic process. Participants reported a variety of emotions during this process that were also stratified into two themes; 1) Feelings towards the Health Professional involved and 2) Internal feelings. During the diagnostic process, they most commonly felt validated/relieved that they were being listened to by a healthcare professional (n = 70; 47.9%) but a large number of participants reported feeling stressed/anxious/overwhelmed about their diagnosis and what that might mean for their future (n = 66; 45.2%). See Table 10 for full details.

**Table 7. How participants felt after being misdiagnosed.**

| Feeling after being misdiagnosed | n | (%) |
|---|---|---|
| **Theme 1: Feelings towards the health professional** | | |
| **Codes:** | | |
| Frustrated/agitated/annoyed/angry | 50 | 40.0 |
| Ignored/invisible/dismissed/not listened to or taken seriously/not seen/disempowered | 30 | 24.0 |
| Not believed/told 'all in my head'/treated like a hypochondriac/like I was going crazy | 26 | 20.8 |
| Didn't agree with health professionals diagnosis/knew there was something more to symptoms | 10 | 8.0 |
| Misunderstood/hurt/let down | 6 | 4.8 |
| **Theme 2: Internal feelings** | | |
| **Codes:** | | |
| Hopeless/devastated/defeated/despairing/disheartened | 25 | 20.0 |
| Upset/sad/tearful/depressed | 17 | 13.6 |
| Confused | 12 | 9.6 |
| Judged | 10 | 8.0 |
| Lonely/abandoned/isolated/marginalised/that no one cared | 5 | 4.0 |
| Scared/anxious | 3 | 2.4 |
| Overwhelmed/stuck/didn't know what to do/feel | 3 | 2.4 |
| Terrible/horrible | 2 | 1.6 |
| Like a failure/weak | 2 | 1.6 |
| Vulnerable | 1 | 0.8 |
| Suicidal | 1 | 0.8 |

Note: Participants frequently talked about more than one emotion.

**Table 8. Actions of participants after misdiagnosis.**

| Action after receiving a misdiagnosis | n | (%) |
|---|---|---|
| Sought self-diagnosis online | 114 | 75.5 |
| Did not seek self-diagnosis online | 37 | 24.5 |
| **Was the online information helpful?** | **n** | **(%)** |
| Yes | 104 | 91.2 |
| No | 10 | 8.8 |
| **Helpful media sources*** | **n** | **(%)** |
| Websites | 88 | 84.6 |
| Facebook support groups | 78 | 75.0 |
| Medical journal articles | 76 | 73.1 |
| YouTube lectures/videos | 37 | 35.6 |
| Podcasts | 15 | 14.4 |
| Others | 13 | 12.5 |

* Participants could select more than one media source.

Ninety-eight percent of participants reported that one or more health professionals treated them well during the diagnostic process with most respondents reporting feeling listened to and validated. Participants describe their feelings during and after their EDS diagnosis:

'I was so overwhelmed that I cried. Not because of the actual diagnosis but because after all these years someone actually listened to me and cared.'

(Participant 16)

'They believed me! It was so unusual and unexpected that I cried with gratitude and relief.' (Participant 28)

**Table 9. Formal diagnostic process.**

| Health Professional responsible for diagnosis of EDS* | n | (%) |
|---|---|---|
| Rheumatologist | 83 | 54.6 |
| Geneticist | 45 | 29.6 |
| General Practitioner (GP) | 13 | 8.5 |
| Other | 12 | 7.9 |
| Physiotherapist | 10 | 6.6 |
| Cardiologist | 4 | 2.6 |
| Sports Physician | 2 | 1.3 |
| **Diagnostic tools used** | **n** | **(%)** |
| Beighton scale | 136 | 89.5 |
| Self-report of pain/other symptoms scales | 117 | 77.0 |
| Clinical criteria | 116 | 76.3 |
| International diagnostic checklist for hEDS | 93 | 61.2 |
| Tools to assess the presence of other potentially associated disorders | 88 | 57.9 |
| Other | 29 | 19.1 |
| Don't know | 6 | 3.9 |

* Some participants indicated that more than one health professional diagnosed their EDS.
* Some participants indicated more than one diagnostic tool.

**Table 10. How participants felt during the diagnostic process.**

| Feelings during and after the Diagnostic process | n | (%) |
|---|---|---|
| **Theme 1: Feelings towards the health professional** | | |
| **Codes:** | | |
| Relieved that a health professional was listening | 70 | 47.9 |
| Like the specialist didn't know enough | 2 | 1.3 |
| **Theme 2: Internal Emotions** | | |
| **Codes:** | | |
| Nervous/anxious/fearful/worried/distressed/concerned about diagnosis | 66 | 45.2 |
| Tired/exhausted | 8 | 5.5 |
| Annoyed/frustrated | 7 | 4.8 |
| Shocked/bemused/confused | 6 | 4.1 |
| Felt that my body had failed me | 2 | 1.4 |
| Alienated/alone | 2 | 1.4 |
| Curious | 2 | 1.4 |
| Less confused | 1 | 0.7 |
| Determined | 1 | 0.7 |

Note: Participants frequently talked about more than one emotion.

'They believed me and didn't think I was exaggerating my symptoms or health issues and weren't dismissive.'

(Participant 17)

'Happy to have a name for something and a reason behind all my pain, but worried that this isn't something that can be cured, and my life is now forever changed.'

(Participant 4)

'Relief and validation. Also, hopelessness and exhaustion for a period after; I realised I had invested all of my energy into working out and proving what was wrong.'

(Participant 30)

## Looking to the future

One hundred and forty-five participants (95.4%) identified that they had to educate health professionals about EDS. This included 48 of the 145 participants (33.1%) feeling like they had to educate 'most or all health professionals' involved in their care. A further 64 participants reported educating their general practitioner (GP) (44.1) about their diagnosis. See Table 11 for further information.

One hundred and forty-six participants (96.1%) provided suggestions for what they thought might improve the diagnostic process for women with EDS living in Australia. Ninety-eight participants (67%) suggested more training, education, and greater awareness for health professionals. Nearly a third of these respondents (n = 42; 28.8%) suggested that, if health professionals listen to and believe their patients and avoid dismissing them, their journey to diagnosis would be better. See Table 12 for further information.

**Table 11. Participant's views regarding educating health professionals about EDS.**

| Did you feel you had to educate health professionals about EDS? | n | (%) |
|---|---|---|
| Yes | 145 | 95.4 |
| No | 7 | 4.6 |
| **If yes, which health professionals?*** | **n** | **(%)** |
| General practitioner | 64 | 44.1 |
| Most health professionals | 27 | 18.6 |
| Physiotherapist | 24 | 16.6 |
| All health professionals | 21 | 14.5 |
| Emergency department doctors | 13 | 9.0 |
| Cardiologist | 8 | 5.5 |
| Nurses | 7 | 4.8 |
| Gastro specialists | 6 | 4.1 |
| Dentist | 4 | 2.8 |
| Neurosurgeon/Neurologist | 4 | 2.8 |
| Rheumatologist | 3 | 2.1 |
| Orthopaedic surgeon | 3 | 2.1 |
| Psychologist | 3 | 2.1 |
| Anaesthetist | 2 | 1.4 |
| Gynaecologist | 2 | 1.4 |
| Exercise physiologist | 1 | 0.7 |
| Spinal surgeon | 1 | 0.7 |
| Allergist | 1 | 0.7 |
| Respiratory physician | 1 | 0.7 |
| Osteopath | 1 | 0.7 |
| Endocrinologist | 1 | 0.7 |
| Pulmonologist | 1 | 0.7 |
| Dietitian | 1 | 0.7 |
| Occupational therapist | 1 | 0.7 |

* Participants could choose more than one option.

One hundred and thirty-four respondents (88.2%) provided suggestions about how information might be better distributed to treating health professionals and/or the broader community to increase awareness of EDS. Some suggestions included more training specifically to doctors and medical students (n = 26; 19.4%) and ongoing training/information sessions available to any health professionals (n = 45; 33.6%), better awareness/campaigns, physical challenges (e.g. a fundraising walk/run), public media coverage (n = 22; 16.4%), more medical conferences (n = 12; 9%) and medical journal articles/research into EDS (n = 15; 11.2%). See Table 13 for further information.

## Discussion

This survey demonstrated that the majority of women with a diagnosis of EDS in Australia experience negative emotions, distress and trauma during their journey to diagnosis, consistent with findings from similar studies from other developed countries. Participants indicated that their journey to diagnosis was long, with the majority of participants in this study first noticing symptoms more than 15 years prior to an EDS diagnosis. This long journey to diagnosis is consistent with previous publications [13,14,17,18,43] and in fact is even longer than

**Table 12. Participants' opinions regarding what might improve the journey to diagnosis of EDS for women living in Australia.**

| Suggestions to improve the journey to diagnosis of EDS* | n | (%) |
|---|---|---|
| Training/education/awareness for health professionals | 98 | 67.1 |
| Health professionals listen to and believe their patients/less dismissal | 42 | 28.8 |
| Increasing public awareness/reducing the stigma | 15 | 10.3 |
| One central service to go to/more EDS specialists | 12 | 8.2 |
| More support for doctors/better resources/more research | 9 | 6.2 |
| Open-minded health professionals/ HPs determined to find answers | 4 | 2.7 |
| Doctors to lose their egos and say they don't know | 4 | 2.7 |
| Early diagnosis/screening | 3 | 2.1 |
| Educating medical students | 3 | 2.1 |
| Clear diagnostic pathway | 3 | 2.1 |
| An overall review of being dismissive of women's experiences | 2 | 1.4 |
| Less judgement of patients | 1 | 0.7 |
| AHPRA to investigate patient complaints | 1 | 0.7 |
| Doctors who do what they say they will do | 1 | 0.7 |

*Participants could choose more than one option.

**Table 13. How might information be best disseminated to increase awareness of EDS?.**

| | n | (%) |
|---|---|---|
| Ongoing training/workshops/info sessions to any medical professional | 45 | 33.6 |
| Taught at med school/university | 26 | 19.4 |
| Awareness campaigns/events/media | 23 | 17.2 |
| Unsure | 18 | 13.4 |
| Medical journal articles including case presentations/real life patient stories/ research/funding into EDS | 15 | 11.2 |
| Medical conferences | 12 | 9.0 |
| Australian foundation | 5 | 3.7 |
| Fact sheets/easy-read flyers/diagnostic pathway | 3 | 2.2 |
| ECHO program | 3 | 2.2 |
| RACGPs | 3 | 2.2 |
| Representation on TV | 3 | 2.2 |
| Contact with advocates | 3 | 2.2 |
| EDS website | 2 | 1.5 |
| Patient led | 1 | 0.7 |
| Incentive to include EDS treatment in public hospital system | 1 | 0.7 |
| Places to go and learn more | 1 | 0.7 |
| TED talks | 1 | 0.7 |
| All specialists in one place | 1 | 0.7 |
| Information distribution | 1 | 0.7 |
| Through professional networks | 1 | 0.7 |
| Newsletters | 1 | 0.7 |
| Books | 1 | 0.7 |
| Add to curriculum in schools | 1 | 0.7 |
| Bulletins from Department of Health | 1 | 0.7 |

*Participants could chose more than one option.

previous reports in this female-only study population. While the majority of participants in this study were diagnosed since 2020, most of them reported that they had been experiencing symptoms for 15 years prior to their official diagnosis. Interestingly, having a relative with a diagnosis of EDS, and therefore family history, did not reduce the time to diagnosis for the participants in this study.

These findings suggest that, similar to other countries [12–14], there has been minimal, if any, increase in awareness of EDS in GPs and other health professionals in Australia in recent years. As GPs have an essential role in the coordination of referrals, especially in patients with complex chronic conditions [44], it is important that education for GPs is prioritised.

The International Diagnostic checklist (IDC) for hEDS, developed by the International Consortium 2017 [45,46], has been established to increase consistency in the diagnosis of hypermobile EDS (hEDS) [47,48], the type of EDS diagnosed in over 90% of the participants in this study. This checklist includes a Beighton score alongside two sets of assessment checklists, which include family history [10,35,43,49]. The presence of all three criteria indicates a clinical diagnosis of hEDS [10,35,43,49]. A GP toolkit is also available through Elhers Danlos Support UK to further guide GP referrals [50]. In this study, participants reported that diagnosing clinicians based their diagnosis of EDS most frequently on a variety of tools including the Beighton scale, clinical criteria and patient-reported pain scores, with only around sixty percent of them indicating that their diagnosis was based on the IDC. Additionally, despite family history being integrated into the IDC, participants with a family history of EDS were not diagnosed faster than those without a family history. However, it must also be acknowledged that patients are not experts in diagnostic testing and therefore the accuracy of this information may not be reliable and cannot be verified.

Most participants felt that they were misdiagnosed prior to receiving their formal diagnosis of EDS with over ten percent of respondents reporting that they received more than nine alternative diagnoses prior to their EDS diagnosis. While patients with hEDS frequently present with comorbidities like migraine headaches, postural orthostatic tachycardia syndrome (POTS) and fibromyalgia (FM) [19], the number of alternative diagnoses reported in this study is higher than those reported in similar studies [19,49,51,52]. In this study 18% of women reported being diagnosed with FM prior to their formal EDS diagnosis. However, as FM alone is not formally recognised by the National Disability Insurance Agency in Australia (NDIA) or Centrelink (Australian welfare system) [53], there is much debate about how helpful a diagnosis of FM is in terms of support and care for people living in Australia.

While some participants were diagnosed with other physical conditions prior to their EDS diagnosis, the majority were diagnosed with psychological conditions, with some participants describing feeling that health professionals perceived them as 'crazy', with symptoms being 'all in their head', or being hypochondriacs. Unfortunately, this finding is consistent with previous research findings [12–14,15,29,54], resulting in an experience which can be dehumanising, causing psychological distress to patients [55] and increasing mistrust in the medical profession [56]. Feeling that they are receiving repeat misdiagnoses may also cause people with EDS to give an inaccurate report of their symptoms in an effort to be heard [57] or can cause them to disengage with health professionals, deterring them from presenting at a hospital for treatment [58,59].

Ehler Danlos Support UK suggest that key roles of the GP should be in diagnosing hEDS, validating the patient's symptoms and co-ordinating care [50]. Therefore, these findings warrant further research to evaluate GP's current knowledge of the IDC and how it is currently being utilised in practice in Australia. This may also reveal barriers to GP referrals which are currently unknown. In previous studies, GPs have cited important factors influencing the referral process as being their internal training, guidelines provided, access to specialists and

confidence in the specialists [44]. Future research should focus on the impact of educating GPs to aid the referral process for suspected EDS, ultimately with the goal of diagnosing this condition earlier.

Many participants reported that they did not feel listened to by their GP and other health professionals and that these professionals were not well informed about EDS or the possibility that they might have EDS. Participants indicated that if health professionals were more open to patients' suggestions, they may have felt more supported during the diagnostic process. This finding is consistent with previous research [12–14,60,61] which also highlights the importance of including patients in open discussion and decision-making processes. Participants viewed it favourably when health professionals listened to them and did not dismiss their concerns. In a study by Bradshaw et al. [62], patients with chronic and complex care needs described being listened to as one of the most important aspects of building trust with their clinician. It is important that health professionals prioritise listening to their patients and contribute to validating their patients' experiences. It is worth noting that most respondents conducted their own research into their symptomatology, usually online. This is consistent with Knight et al's study [46] which reported that people with an EDS diagnosis are not only aware of current EDS research but also of the deficiencies within the EDS care pathway. This further highlights the lack of support in place for patients as they wait for a formal diagnosis of EDS.

Physiotherapy has a key role to play in the management of hypermobile EDS [63]. A 2011 survey by Rombaut et al. found that 63.4% of the hypermobile EDS patients enrolled in a physical therapy program reported a positive effect of the treatment. However, a recent study showed a lack of confidence by physiotherapists in assessment and management of hypermobile EDS [63]. Physiotherapists in this study were reported to be central to the diagnosis of EDS in 6.6% of cases, emphasising their potential role in identifying potential EDS cases and making appropriate referrals to confirm this diagnosis. Given the importance of their role in EDS rehabilitation in Australia, EDS education for physiotherapists should also be prioritised and integrated into undergraduate education as a mandated component of the curriculum. This education would include review of the components of the IDC alongside theoretical and practical learning regarding the appropriate treatment for patients with all types of EDS.

Despite a geneticist diagnosing almost a third of EDS cases in this study, there is currently no identifable gene which has been found to be associated with hEDS [46,47], the type of EDS diagnosed in the majority of patients [8,9]. Genetic testing is available for every type of EDS except for hEDS [8,9]. This suggests that, while geneticists have strong knowledge of all types of EDS, the services that they provide are mainly suitable for only a small number of EDS patients. Subsequently, referral pathways from GPs for people with suspected EDS need to be more clearly guided to ensure that resources are been utilised optimally.

Ultimately, participants in this study suggested that more education and training for all health professionals in Australia, as well as raising public awareness of EDS, may help to improve the diagnostic journey for individuals living with EDS in the future.

## Limitations

As with the majority of studies, the design of the current study is subject to limitations.

One such limitation was the method of recruitment through convenience sampling. The survey was only offered and advertised online on two social media platforms. This prevented access for participants who might not be active online, not have access to smart technology or may not engage in social media. Due to self-selection, there may be differences between the people who choose to participate in the survey compared to those who choose not to

participate [64]. Additionally, the survey was only open for six weeks which may not have given some participants adequate time to find out about the survey and/or complete it. The convenience sampling method used in the study is prone to biases that cannot be generalised beyond the specific sample studied [65]. As this study was a pilot study, the results are helpful in generating hypotheses for more rigorous research designs in the future.

While the survey content validity was increased through discussions between the two researchers and with women with EDS living in Brisbane, Australia [33], the questionnaire utilised in this study was not a validated measure and therefore its psychometric properties and its methodological quality have not been evaluated [66]. The wording of Question 9 regarding 'misdiagnosis' could be perceived as leading. Patients' perceptions of being misdiagnosed, and their understanding of what diagnostic tool has been used, can be highly subjective and influenced by their understanding of their symptoms and conditions. The questionnaire does not include a mechanism for verifying the reported misdiagnoses against medical records or clinical evaluations. Without validation from healthcare professionals, it is impossible to determine if the reported misdiagnoses are accurate. While the lead researcher's expertise is in managing EDS complications, including an expert in EDS diagnosis, such as a geneticist or rheumatologist, would have increased the questionnaire's accuracy and relevance.

Some participants who were diagnosed some years ago may not recall details of their diagnosis accurately leading to risk of bias; this may either underestimate or overestimate the issues raised in the survey [67].

Focusing only on the lived experience of women in Australia is another limitation of this study. The lived experiences of all genders, including men, non-binary and gender fluid individuals, is under-researched.

In the future, it would be beneficial to recruit more broadly using a variety of communication methods to reach more individuals with EDS, include all genders, and delve deeper into the lived experiences of all people living with EDS in Australia and beyond.

## Conclusion

Based on these findings, it is reasonable to propose that increased education of GPs and other key health professionals involved in EDS care, could improve the journey to diagnosis for women living in Australia. Increasing awareness in the general public as well as within the health and disability sectors may allow for quicker identification of EDS symptoms, reducing unnecessary patient stress and distress during the diagnostic process and lead to shorter lengths of time to diagnosis of this rare condition. This could mean better mental health and wellbeing for patients before, during and after their diagnosis, which may lead to better health outcomes as they adjust to living with EDS. Providing education and training to GPs and other health professionals will be challenging but may be essential to improve the lives of people with EDS in Australia.

## Supporting information

**S1 Appendix. Participant survey.**
(DOCX)

## Acknowledgments

We would like to thank all of the people with EDS who participated in the design of this research study and also those who completed the survey.

## Author Contributions

**Conceptualization:** Rachel Trudgian.

**Data curation:** Rachel Trudgian.

**Formal analysis:** Rachel Trudgian, Terri Flood.

**Investigation:** Rachel Trudgian.

**Methodology:** Rachel Trudgian, Terri Flood.

**Project administration:** Rachel Trudgian, Terri Flood.

**Resources:** Rachel Trudgian.

**Software:** Terri Flood.

**Supervision:** Terri Flood.

**Validation:** Terri Flood.

**Visualization:** Rachel Trudgian.

**Writing – original draft:** Rachel Trudgian.

**Writing – review & editing:** Terri Flood.

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
