## [Decision Letter · Decision Letter 0]

12 Jun 2024

PONE-D-24-18819An exploration of the journey to diagnosis of Ehlers-Danlos Syndrome (EDS) for women living in AustraliaPLOS ONE

Dear Dr. Flood,

Thank you for submitting your manuscript to PLOS ONE. After careful consideration, we feel that it has merit but does not fully meet PLOS ONE’s publication criteria as it currently stands. Therefore, we invite you to submit a revised version of the manuscript that addresses the points raised during the review process.

We look forward to receiving your revised manuscript.

Kind regards,

Martin E. Matsumura, MD

Academic Editor

PLOS ONE

3. For studies involving third-party data, we encourage authors to share any data specific to their analyses that they can legally distribute. PLOS recognizes, however, that authors may be using third-party data they do not have the rights to share. When third-party data cannot be publicly shared, authors must provide all information necessary for interested researchers to apply to gain access to the data. (https://journals.plos.org/plosone/s/data-availability#loc-acceptable-data-access-restrictions) 

Additional Editor Comments 

Please address all feedback below.

Please note that 1 reviewer feedback is on an attached document which can be accessed below

Editor note: I feel that the inclusion of variables with "zero" respondents (comment regarding table 2) can be left in the tables as these reflect the questions asked in the survey.  

Reviewers' comments:

Reviewer's Responses to Questions

**Comments to the Author**

1. Is the manuscript technically sound, and do the data support the conclusions?

Reviewer #1: Yes

Reviewer #2: Partly

2. Has the statistical analysis been performed appropriately and rigorously? 

Reviewer #1: Yes

Reviewer #2: N/A

3. Have the authors made all data underlying the findings in their manuscript fully available?

Reviewer #1: No

Reviewer #2: Yes

4. Is the manuscript presented in an intelligible fashion and written in standard English?

Reviewer #1: Yes

Reviewer #2: Yes

5. Review Comments to the Author

Reviewer #1: I have several minor issues to address on revision. Details are provided in attached document. Primary issues are related to table formatting and use of the term "misdiagnosis" to designate earlier diagnoses.

Reviewer #2: A detailed review of the manuscript titled: “An exploration of the journey to diagnosis of Ehlers-Danlos Syndrome (EDS) for women living in Australia.”

There are several critical issues that require attention for the study to achieve its intended impact. The authors should revise the manuscript and address these major critics. Below are my comments and suggestions:

1-Authors grouped all Ehlers-Danlos syndrome (EDS) subtypes together, and provided a prevalence. The method used estimated the prevalence of EDS globally and applies it to the Australian population. Their statistical assessment based on a presumed and inaccurate prevalence, considering the variable and largely unknown prevalence of individual EDS subtypes.

The study specifically addresses the prevalence and clinical characteristics of hypermobile Ehlers-Danlos Syndrome (hEDS). Given the distinct nature and higher prevalence of hEDS compared to other EDS subtypes, it is imperative to focus exclusively on hEDS to provide accurate and clinically relevant data.

2- The convenience sampling method used in the study is prone to biases that cannot be generalized beyond the specific sample studied. The authors should present their data as an exploratory or pilot study, generating hypotheses for more rigorous research designs using probability sampling methods. This approach would prevent the overstatement of findings and acknowledge the limitations inherent to convenience sampling.

3- The authors should explicitly include self-selection bias, inaccurate or false data, and recall bias as limitations of the study. Recognizing these biases will provide a more balanced interpretation of the study’s findings.

4- Several statements in the article represent inaccurate information about hEDS, including:

o “EDS alters collagen structures in the body leading to multi-systemic complications including generalized joint hypermobility,…” and “functional decline is common place.” The etiology of hEDS is unknown, and it is generally not caused by defective collagen. The condition is not progressive.

o “Several reports indicate that over 70% of EDS cases occur in women with over 90% of hEDS cases being female.” While hEDS is more common in females, there is no sex predominance in other EDS subtypes.

o “Due to the ‘invisible’ nature of EDS, patient reports of their symptoms, particularly before they are accurately diagnosed with EDS, are often dismissed.” The term "invisible" is not appropriate; the authors likely refer to the lack of a pathognomonic biomarker. Additionally, this statement refer specifically to hEDS and should be clarified.

5-The initial questionnaire design was based on an extensive literature search and the lead researcher’s anecdotal experience as an occupational therapist. While practical insights are valuable, the lead researcher’s expertise is in managing EDS complications rather than diagnosing it. Including an expert in EDS diagnosis, such as a geneticist or rheumatologist, would’ve ensured the questionnaire's accuracy and relevance. The authors should detail the specific expertise of the second researcher, clearly outlining their qualifications and experience in relation to EDS.

6- Revision of all these statements are required:

“The majority of participants (n=129; 84.9%) were misdiagnosed before being correctly diagnosed with EDS. Most participants were diagnosed with anxiety and/or depression (n=91; 73.4%) or another psychological condition (n=47; 37.9%), chronic pain (n=67; 54%), myalgic encephalomyelitis/chronic fatigue syndrome (ME/CFS) (n=58; 46.8%), or another physical condition including fibromyalgia (n=18; 17.7%). Over 10% of respondents (n=14) received more than nine misdiagnoses before being correctly diagnosed with EDS.”

“Most participants were misdiagnosed prior to receiving their formal diagnosis of EDS with over ten percent of respondents receiving more than nine misdiagnoses.”

“Participants consistently reported being referred to numerous specialists who lack knowledge regarding EDS, often resulting in misdiagnosis.”

“While misdiagnosis is expected and consistent with previous findings the extent of misdiagnosis reported in this study is astonishing.”

These statements are stemmed from a question that does not specify the timeframe, context, or criteria for what constitutes a "misdiagnosis (questions 11, and 16. This lack of clarity can lead to inconsistent interpretations and responses from participants. Without rigorous validation and a clear, objective framework for assessing misdiagnoses, self-reported data on this topic are likely to be inaccurate and unreliable, making all the conclusions unreliable.

Patients' perceptions of being misdiagnosed, and their understanding of what diagnostic tool has been used, can be highly subjective and influenced by their understanding of their symptoms and conditions. The question does not include a mechanism for verifying the reported misdiagnoses against medical records or clinical evaluations. Without validation from healthcare professionals, it is impossible to determine if the reported misdiagnoses are accurate.

Conditions such as Chronic Fatigue Syndrome (CFS), Myalgic Encephalomyelitis (ME), and fibromyalgia have evolving diagnostic criteria, which can vary between practitioners and over time. In addition, patients who feel dissatisfied with their care or outcomes may be more likely to report being misdiagnosed, introducing a potential bias in the survey responses.

7-The statement, "As FM alone is not formally recognized by the National Disability Insurance Agency in Australia (NDIA) or Centrelink (Australian welfare system), there is much debate about how helpful a diagnosis of FM is and whether FM is diagnosed in the absence of an alternative clear diagnosis," requires revision for several reasons:

Fibromyalgia (FM) is recognized globally and has been assigned an ICD code (International Classification of Diseases) long before hypermobile Ehlers-Danlos Syndrome (hEDS). The ICD-10 code for FM is M79.7, and it has been widely used in medical diagnosis and research. This recognition reflects the consensus within the medical community about FM as a legitimate and distinct medical condition.

The American College of Rheumatology has established specific diagnostic criteria that healthcare providers use to diagnose FM based on clinical findings. The lack of formal recognition by specific national agencies, such as the NDIA or Centrelink, does not diminish the validity of the diagnosis itself. It is important to distinguish between administrative recognition for the purposes of disability support and the medical validity of the diagnosis. The statement conflates these two distinct issues, potentially misleading readers about the legitimacy of FM as a clinical diagnosis.

In addition FM can co-occur with other conditions. Misrepresenting the status of FM can contribute to stigma and misunderstanding, which can negatively impact patients seeking support and treatment.

Therefore, it is recommended that the statement be revised to reflect the established medical recognition of fibromyalgia and clarify the distinction between administrative recognition for disability support and medical diagnosis. An accurate portrayal will ensure the article is informative and respectful of the condition's legitimacy.

8- The statement, "While Physiotherapists made only 6.6% of the EDS diagnoses of participants in this study, this group of allied health professionals are aptly positioned to diagnose and treat EDS in conjunction with referrals to other appropriate specialists," requires clarification.

While physiotherapists play a crucial role in the multidisciplinary care of patients with hypermobile Ehlers-Danlos Syndrome (hEDS), their primary expertise lies in the management and rehabilitation of musculoskeletal issues, rather than in the diagnosis of complex genetic conditions. According to the 2017 International Classification of the Ehlers-Danlos Syndromes, the diagnosis of hEDS includes the exclusion of alternative diagnoses, which involves a comprehensive clinical assessment that includes detailed patient history, physical examination, and the exclusion of other conditions with overlapping phenotypes. This process typically requires the expertise of a medical professional trained in genetics, rheumatology, or another relevant specialty. Physiotherapists do not possess the necessary training or authority to perform the required comprehensive diagnostic workup.

6. PLOS authors have the option to publish the peer review history of their article (what does this mean?). If published, this will include your full peer review and any attached files.

Reviewer #1: No

Reviewer #2: No

---

## [Editor Report · Decision Letter 1]

9 Jul 2024

An exploration of the journey to diagnosis of Ehlers-Danlos Syndrome (EDS) for women living in Australia

PONE-D-24-18819R1

Dear Dr. Flood,

We’re pleased to inform you that your manuscript has been judged scientifically suitable for publication and will be formally accepted for publication once it meets all outstanding technical requirements.

Kind regards,

Martin E. Matsumura, MD

Academic Editor

PLOS ONE

Additional Editor Comments (optional):

Thank you for addressing all reviewer comments. My only additional comment- the statements you added to tables 11 and 12 to indicate patients could pick multiple answers- please change "chose" to "choose"

---

## [Editor Report · Acceptance letter]

17 Jul 2024

PONE-D-24-18819R1 

PLOS ONE

Dear Dr. Flood, 

I'm pleased to inform you that your manuscript has been deemed suitable for publication in PLOS ONE. Congratulations! Your manuscript is now being handed over to our production team.

Kind regards, 

on behalf of

Dr. Martin E. Matsumura 

Academic Editor

PLOS ONE